# Inductive Cognitive Diagnosis for Fast Student Learning in Web-Based Online Intelligent Education Systems

## ABSTRACT

Cognitive diagnosis aims to gauge students' mastery levels based on their response logs. Serving as a pivotal module in web-based online intelligent education systems (WOIESs), it plays an upstream and fundamental role in downstream tasks like learning item recommendation and computerized adaptive testing. WOIESs are open learning environments where numerous new students constantly register and complete exercises. In WOIESs, efficient cognitive diagnosis is crucial to fast feedback and accelerating student learning. However, the existing cognitive diagnosis methods always employ intrinsically transductive student-specific embeddings, which become slow and costly due to retraining when dealing with new students who are unseen during training. To this end, this paper proposes an inductive cognitive diagnosis model (ICDM) for fast new students' mastery levels inference in WOIESs. Specifically, in ICDM, we propose a novel student-centered graph (SCG). Rather than inferring mastery levels through updating student-specific embedding, we derive the inductive mastery levels as the aggregated outcomes of students' neighbors in SCG. Namely, SCG enables to shift the task from finding the most suitable student-specific embedding that fits the response logs to finding the most suitable representations for different node types in SCG, and the latter is more efficient since it no longer requires retraining. To obtain this representation, ICDM consists of a construction-aggregation-generation-transformation process to learn the final representation of students, exercises and concepts. Extensive experiments across real-world datasets show that, compared with the existing cognitive diagnosis methods that are always transductive, ICDM is much faster while maintains the competitive inference performance for new students.

## CCS CONCEPTS

• **Applied computing** → Education; • **Computing methodologies** → Machine learning.

## KEYWORDS

Cognitive Diagnosis, Web-based Online Intelligent Education Systems, Inductive Learning

## 1 INTRODUCTION

With the proliferation of vast online learning resources and web-based online intelligent educational systems (e.g., KhanAcademy.org, junyiacademy.org), a growing number of students and learners increasingly turn to the web as a primary medium for education. Web-based online intelligent education systems (WOIDSs) [12, 33] enhance personalized student learning through computer-assisted methods, offering a wealth of educational resources (e.g., courses, exercises). Cognitive diagnosis (CD), as the cornerstone of WOIDSs,

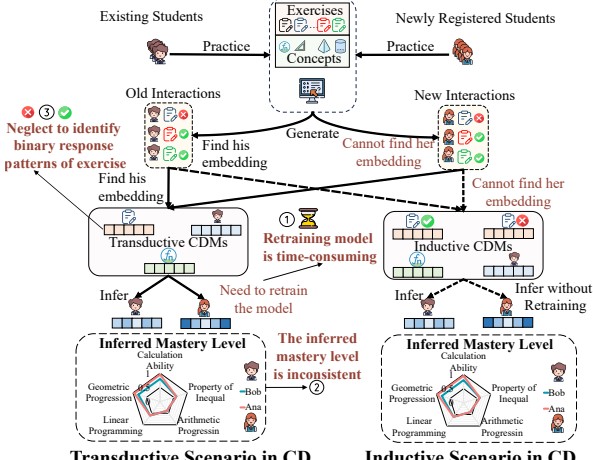

**Web-based Online Intelligent Education Systems**

**Figure 1: Transductive and inductive scenarios in CD.**

plays an upstream and fundamental role in them, affecting downstream modules such as computer adaptive testing [35], course recommendation [32] and learning path suggestions [15], etc. Specifically, by analyzing students' historical response logs, CD endeavors to infer students' underlying mastery levels (Mas) and shed light on attributes of exercises (e.g., difficulty, discrimination).

In recent years, a plenty of cognitive diagnosis models (CDMs) have come to the fore, like item response theory (IRT) [8] and neural cognitive diagnosis odel (NCDM) [25]. IRT employs a latent factor to represent Mas, using a simple logistic function as the interaction function (IF). NCDM replaces the traditionally IF with multi-layer perceptrons (MLPs) and adopts concept-specific (i.e., set the embedding dimension as the number of concepts) vectors to depict Mas. As embedding-based methods continue to advance rapidly and become mainstream, researchers are showing a growing preference for converting both students and exercises into vectorized forms, further refining them with various techniques [6, 14, 17, 26]. Notably, most existing CDMs employ intrinsically transductive student-specific embeddings and thus have to access the response logs of all students during the training phase.

Existing WOIDSs are open learning environment where a vast number of new students register and complete a multitude of exercises, as shown in the right part of Figure 1. It means that the number of students is uncertain and cannot be pre-defined. In WOIDSs, students expect to obtain immediate and timely feedback on their diagnostic results (i.e., mastery levels), which can fast aid in their self-improvement or assist teachers in providing tailored advice. Unfortunately, most existing CDMs are transductive and could struggle to provide such diagnostic results quickly due to

their reliance on student-specific embedding and their neglect of the identifying binary response patterns. These two factors indicate that they need to be retrained to infer new students' Mas, which is time-consuming and unacceptable in practice. Thus, inductive cognitive diagnosis is more suitable for WOIDSs under open learning environment. In the inductive scenarios of CD, there is an urgent need for efficient and interpretable approach capable of inferring the Mas of new students without model retraining. For inductive cognitive diagnosis, the only related work is incremental CD (ICD) [22] which targets streaming log data in CD. In [22], it aims to update the Mas at the next moment without retraining, which is different from the goal of focusing on inferring new students' Mas.

Actually, inductive cognitive diagnosis for fast inferring Mas of newly registered students is non-trivial. In most cases, we only have access to students' response logs. It's not feasible to establish a connection between new and existing students based on their background characteristics (e.g., learning environment, parents' education level) to deduce the new students' Mas. The sole information at our disposal is the response logs from both new and existing students. Moreover, current CDMs overlook the consistency of Mas. Owing to the random initialization of parameters, the Mas they infer often display inconsistency. That is, students with identical response logs could exhibit varied Mas. This is unreasonable, as we lack knowledge about the students' individual background information.

To this end, this paper formally defines the inductive scenario in CD and proposes an inductive cognitive diagnosis model (ICDM) for fast new students' mastery levels inference in WOIESs. To be specific, in ICDM we propose a novel student-centered graph (SCG). Rather than inferring mastery levels through updating student-specific embedding, we derive the inductive mastery levels as the aggregated outcomes of students' neighbors in SCG. That is to say, SCG enables to shift the task from finding the most suitable student-specific embedding that fits the response logs to finding the most suitable representations for different node types in SCG, and the latter is more efficient since it no longer requires retraining. To achieve this representation, ICDM consists of a construction-aggregation-generation-transformation (CAGT) process to learn the final representation of students, exercises and concepts which can be seamlessly integrated into various IFs. Moreover, we also design a novel global-level IF to predict students' performance on exercises. Extensive experiments across real-world datasets show that, compared with the existing CDMs that are always transductive, ICDM is much faster while maintains the competitive inference performance for new students.

The subsequent sections respectively recap the related work, present the preliminaries, introduce the proposed ICDM, show the empirical analysis and conclude the paper.

## 2 RELATED WORK

**Cognitive Diagnosis.** CDMs are used to evaluate student profiles by employing either latent factor models, such as IRT [16] and multidimensional IRT (MIRT) [20], or models based on patterns of concept mastery, such as deterministic input, noisy and gate model (DINA) [3]. For instance, DINA, a typical example of CDMs, utilizes binary independent variables to represent mastery states, where 0 indicates an unmastered state and 1 represents a mastered state. Favored by recent deep learning techniques, researchers achieve great success in large-scale interactions circumstances. NCDM [25] employs MLPs as IF and represents mastery patterns as continuous variables within the range of $[0, 1]$. Various approaches have been employed to capture fruitful information in the response logs, such as MLP-based [17, 26], graph attention network based [6], Bayesian network based [14]. However, existing CDMs are tailored for the transductive scenario in CD and cannot be directly applied to the inductive scenario. The only related CDM is ICD [22] which targets streaming log data. Its goal is to update the Mas at the next moment without retraining. However, in WOIESs where new students often generate vast amounts of response data, using such an approach can become prohibitively costly due to frequent updates.

**Inductive Matrix Completion for Collaborative Filtering.** Given that contemporary CDMs do not possess inductive learning abilities, and factoring in that they are evaluated by predicting student outcomes on new, unattempted exercises (similar to filling out the rating matrix), and only having IDs as distinguishing features for students, exercises and concepts, we turn to the methods of Inductive Matrix Completion for Collaborative Filtering [10, 19, 30, 31, 34] to draw comparisons and encapsulate pertinent studies. Nevertheless, these approaches either demand significant computational resources [10, 30], potentially hindering timely feedback for a vast number of students in WOIDSs, or they may compromise on accuracy [19, 34]. INMO [31], a state-of-the-art approach, presents a model-agnostic inductive collaborative filtering methodology that adeptly chooses template users and items grounded in thorough theoretical analysis. However, INMO's theory is constructed on the premise that the IF is dot product-based, making it unsuitable for the CD. This is because the IFs in CD need to adhere to a psychometric assumption (i.e., monotonicity assumption). Hence, directly applying methods from Inductive Matrix Completion for Collaborative Filtering is impractical. Not only is it time-consuming, but it also fails to capitalize on the intricate student-exercise-concept relationships or adhere some assumptions inherent in CD.

## 3 PRELIMINARIES

This section first introduces the fundamental elements of cognitive diagnosis. Subsequently, we formalize both the transductive and inductive cognitive diagnosis tasks.

**Cognitive Diagnosis Basis.** Consider web-based online intelligent education systems (WOIDSs) which contain two sets: $E = \{e_1, \ldots, e_M\}$, and $C = \{c_1, \ldots, c_Z\}$. They symbolize exercises and knowledge concepts, with respective sizes of $M$ and $Z$. $\mathbf{Q}$ represents the relationship between exercises and knowledge concepts, which can be regarded as a binary matrix $\mathbf{Q} = (\mathbf{Q}_{ij})_{M \times Z}$, where $\mathbf{Q}_{ij} \in \{0, 1\}$ means whether $e_i$ relates to $c_j$ or not. Here, we assume that both the exercises and concepts are static, implying that their quantities remain constant. Students in set $S = \{s_1, \ldots, \}$, driven by unique interests and requirements, select exercises from $E$. The results are documented as response logs. Specifically, these logs can be illustrated as triplets $T = \{(s, e, r) | s \in S, e \in E, r_{se} \in \{0, 1\}\}$. $r_{se} = 1$ represents correct and $r_{se} = 0$ represents wrong. In this paper, we treat response logs as rating matrix $\mathbf{R} \in \mathbb{R}^{|S| \times M}$ where $|S|$ denotes the size of set $S$. It contains three categorical values (1

means right, 0 means no interaction and −1 means wrong. Cognitive Diagnosis is to infer $\mathbf{Mas} \in \mathbb{R}^{|S| \times Z}$ which denotes the latent Mas of students on each knowledge concept based on $\mathbf{R}$.

**Transductive Cognitive Diagnosis.** Current CDMs assess student performance within a transductive scenario as shown in the left part of Figure 1. Formally, given the students' set $|S|$, a rating matrix $\mathbf{R} \in \mathbb{R}^{|S| \times M}$, a binary matrix $\mathbf{Q}$, our goal is to infer $\mathbf{Mas} \in \mathbb{R}^{|S| \times Z}$, which denotes the latent Mas of all students.

**Inductive Cognitive Diagnosis.** The frequent registration and participation of new students in the WOIDSs can be characterized as an inductive scenario. As illustrated in the right part of Figure 1, it expects the CDMs to accurately diagnose for newcomers *without retraining the models*. Formally, given the existing students' set $S^O$, unseen students' set $S^U$ where $S^O \cap S^U = \emptyset, S^O \cup S^U = S, |S^O| = N^O, |S^U| = N^U$, rating matrices $\mathbf{R}^O, \mathbf{R}^U$ and a binary matrix $\mathbf{Q}$. The goal is to infer $\mathbf{Mas}^U \in \mathbb{R}^{N^U \times Z}$, which denotes the latent Mas of new students on each concept.

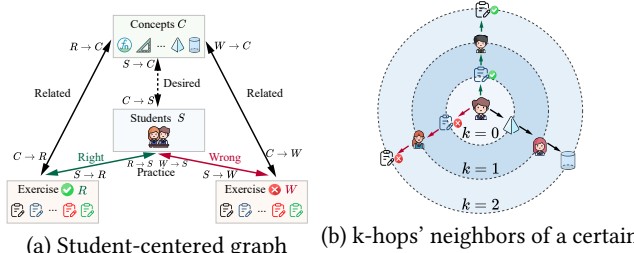

(a) Student-centered graph   (b) k-hops' neighbors of a certain student

**Figure 2: The proposed student-centered graph (SCG).**

## 4 METHODOLOGY: THE PROPOSED ICDM

This section begins by presenting the innovative student-centered graph. Following that, we delve into the CAGT process, which allows us to derive representations for students, exercises, and concepts. Subsequently, the proposed global-level interactive function (GLIF) is introduced. We conclude the section by discussing the model's training. Notably, the strength of ICDM lies in addressing the inductive scenario in CD. Hence, all its underlying notions are derived from this scenario. Nevertheless, we claim that ICDM is versatile enough to be applied in the transductive scenario as well. The framework of ICDM is shown in Figure 3.

**Student-Centered Graph.** As illustrated in Figure 2(a), focusing on students, the student-centered graph (SCG), denoted as $\mathcal{G} = (\mathcal{V}, \mathcal{U})$, comprises four types of nodes and edges. $\mathcal{V} = S \cup E_R \cup E_W \cup C$ involves students, exercises with right pattern, exercises with wrong pattern and concepts, $\mathcal{E}$ involves interactions between $S$ and $E_R$ (i.e., "Right"), $S$ and $E_W$ (i.e., "Wrong"), $E_R$ and $C$ (i.e., "Related"), $E_W$ and $C$ (i.e., "Related"), $S$ and $C$ (i.e., "Desired") which will introduced later. For instance, if $u_{s_i, r_j} \in \mathcal{U}$, the $i$-th student practice the $j$-th exercise right.

### 4.1 The CAGT Process

**Construction.** The sole features in CD is response logs and $\mathbf{Q}$. Evidently, it is essential to decompose these intricate logs into their constituent elements: student, exercise with right pattern, exercise

with wrong pattern, and concept. We encode them with trainable embeddings $\mathbf{H}_s \in \mathbb{R}^{N^O \times d}, \mathbf{H}_r \in \mathbb{R}^{M \times d}, \mathbf{H}_w \in \mathbb{R}^{M \times d}, \mathbf{H}_c \in \mathbb{R}^{Z \times d}$. For instance, $h_{s_i} \in \mathbb{R}^{1 \times d}$ denotes the row vector of $i$-th student. Notably, we employ response-aware embeddings to capture the binary response patterns in exercises, patterns that were overlooked by previous CDMs, in order to infer the new students' $\mathbf{Mas}$. Moreover, based on the phenomenon that the exercises chosen by students might potentially reveal their preferences for learning specific knowledge concepts, we construct the involvement matrix $\mathbf{I}^O \in \mathbb{R}^{N^O \times Z}$ with $\mathbf{Q}$ and $\mathbf{R}^O$. For example, if student $s_1$ practices exercise $e_2$ which is associated with concept $c_3$, then the value of $\mathbf{I}^O_{13}$ is set to 1. Finally, we construct the SCG $\mathcal{G}$ based on $\mathbf{R}^O, \mathbf{I}^O$ and $\mathbf{Q}$. Specifically, if $\mathbf{R}^O_{ij} = 1, u_{s_i, r_j} \in \mathcal{U}$; if $\mathbf{R}^O_{ij} = -1, u_{s_i, w_j} \in \mathcal{U}$; if $\mathbf{Q}^O_{jz} = 1, u_{r_j, c_z} \in \mathcal{U} \vee u_{w_j, c_z} \in \mathcal{U}$; if $\mathbf{I}^O_{iz} = 1, u_{s_i, c_z} \in \mathcal{U}$.

**Aggregation.** As demonstrated in Figure 2(b), the k-hop neighbors of a specific student convey rich information. For instance, the one-hop neighbors directly characterize Bob in WOIDSs, while the behaviors of the two-hop neighbors are somewhat similar to Bob's. They might have answered a particular exercise right or wrong. Clearly, aggregating the information from neighbors is instrumental in more adeptly deducing Mas. Analogously, this holds true for the attributes of exercises and concepts. During the Aggregation phase, we aim to aggregate information from different types of neighbors for each specific node type which can be expressed as

$$h_{s_i}^{k+1}(R \to S) = \text{AGG}\left(\text{Drop}^k \left\{ h_{r_j}^k \,|\, \forall j, \text{ if } u_{s_i, r_j} \in \mathcal{U} \right\}\right),$$

$$h_{s_i}^{k+1}(W \to S) = \text{AGG}\left(\text{Drop}^k \left\{ h_{w_j}^k \,|\, \forall j, \text{ if } u_{s_i, w_j} \in \mathcal{U} \right\}\right), \quad (1)$$

$$h_{s_i}^{k+1}(C \to S) = \text{AGG}\left(\text{Drop}^k \left\{ h_{c_z}^k \,|\, \forall z, \text{ if } u_{s_i, c_z} \in \mathcal{U} \right\}\right).$$

The term $h_{s_i}^{k+1}(R \to S), h_{s_i}^{k+1}(W \to S), h_{s_i}^{k+1}(C \to S)$ denotes the aggregated outcome from exercises with right pattern, exercises with wrong pattern and concepts at depth $k + 1$ respectively. AGG stands for the aggregator function, which consolidates the information from the provided vectors of neighbors. Here, the aggregator function can be of various types, such as mean aggregation [9] or graph attention [1, 24] which will be analyzed in experiments. $\text{Drop}^k$ signifies a layer-wise neighbor dropout, indicating that the dropout probability, represented by $p$, calculated as $p = \alpha + \beta k$. As $k$ increases incrementally, the noise (i.e, guess or slip) introduced during the propagation process amplifies, necessitating a larger value for $p$. Introducing dropout serves a dual purpose: it mitigates the effects of the noise and boosts the model's generalizability.

Finally, given that the significance of neighbors from k-hops gradually diminishes as $k$ increases in CD, we employ a descending accumulation method to integrate the aggregated outcomes from different $k$, which can be expressed as

$$h_{s_i}(R \to S) = \sum_{k=0}^{K} \frac{1}{k+1}(h_{s_i}^k(R \to S)),$$

$$h_{s_i}(W \to S) = \sum_{k=0}^{K} \frac{1}{k+1}(h_{s_i}^k(W \to S)), \quad (2)$$

$$h_{s_i}(C \to S) = \sum_{k=0}^{K} \frac{1}{k+1}(h_{s_i}^k(C \to S)),$$

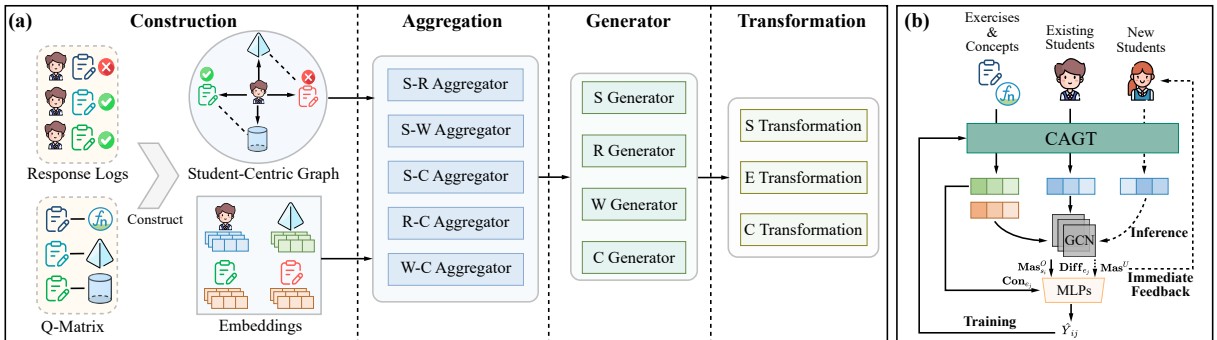

**Figure 3: Framework of the proposed ICDM: (a) The details of CAGT process. (b) Training and inference.**

where $K$ represents the pre-defined maximum depth. Herein, we just utilize the student as a instance. Through analogous computations, we can further deduce $h_{r_j}(S \rightarrow R), h_{r_j}(C \rightarrow R), h_{w_j}(S \rightarrow W), h_{w_j}(C \rightarrow W), h_{c_z}(R \rightarrow C), h_{c_z}(S \rightarrow C)$, and $h_{c_z}(W \rightarrow C)$.

**Generation.** The final representations of students, exercises with right pattern, exercises with wrong pattern and concepts can be generated from the results mentioned above.

$$
\begin{aligned}
h_{s_i} &= \text{GEN}(h_{s_i}(R \rightarrow S), h_{s_i}(W \rightarrow S), h_{s_i}(C \rightarrow S)), \\
h_{r_j} &= \text{GEN}(h_{r_j}(S \rightarrow R), h_{r_j}(C \rightarrow R)), \\
h_{w_j} &= \text{GEN}(h_{w_j}(S \rightarrow W), h_{w_j}(C \rightarrow W)), \\
h_{c_z} &= \text{GEN}(h_{c_z}(R \rightarrow C), h_{c_z}(W \rightarrow C), h_{c_z}(S \rightarrow C)),
\end{aligned} \quad (3)
$$

where $h_{s_i}$ denotes the generated final representation of $i$-th student. GEN represents weighted generator, which we will elaborate on in the subsequent text. Through this approach, we discern that the representation of the $i$-th student is exclusively associated with its k-hops' neighbors, such as exercises with right patterns, exercises with wrong patterns, and the concepts involved. Therefore, when a new student registers and answers exercises in WOIDSs, we can directly infer their Mas based on their performance in answering the exercises and their desired concepts without retraining. Given that different types of neighbor information can be perceived as various views for a specific node (e.g., students), we opt to use a data-driven approach [27] to fuse them together through a weighted summation. Here, we use the student as an example. The weight corresponding to $h_{s_i}(R \rightarrow S)$ can be computed as

$$
w_R = \mathbf{a}_s \tanh \left( h_{s_i}(R \rightarrow S)\mathbf{W}_s^g + \mathbf{b}_s^g \right)^\top , \quad (4)
$$

where $\mathbf{a}_s \in \mathbb{R}^{1 \times d}$ denotes attention vector, $\mathbf{W}_s^g \in \mathbb{R}^{d \times d}$ and $\mathbf{b}_s^g \in \mathbb{R}^{1 \times d}$ are trainable parameters in the student representation generation phase. Based on Eq (4), $w_W, w_C$ can be computed, each representing the weight of the respective part. With the help of Softmax, the normalized weight of $h_{s_i}(R \rightarrow S)$ is $\tilde{w}_R = \frac{e^{w_R}}{e^{w_R}+e^{w_W}+e^{w_C}}$. Similarly, we can derive $\tilde{w}_W, \tilde{w}_C$. Ultimately, the representation of $i$-th student can be expressed as

$$
h_{s_i} = \tilde{w}_R h_{s_i}(R \rightarrow S) + \tilde{w}_W h_{s_i}(W \rightarrow S) + \tilde{w}_C h_{c_z}(C \rightarrow S). \quad (5)
$$

Similarly, following the same procedure, we can derive $h_{r_j}, h_{w_j}, h_{c_z}$.

**Transformation.** Many CDMs [6, 25, 26] are based on the concept-specific pattern, indicating the number of dimensions equals the number of knowledge concepts $Z$. We use simple yet effective linear transformation to change the dimension from $d$ to $Z$.

$$
h_{s_i} = h_{s_i}\mathbf{W}_s^t + b_s^t, \; h_{e_j} = (h_{r_j} \odot h_{w_j})\mathbf{W}_e^t + b_e^t, \; h_{c_z} = h_{c_z}\mathbf{W}_c^t + b_c^t , \quad (6)
$$

where $\odot$ means Hadamard product, $\mathbf{W}_s^t, \mathbf{W}_e^t, \mathbf{W}_c^t \in \mathbb{R}^{d \times Z}, b_s^t, b_e^t, b_c^t \in \mathbb{R}^{1 \times Z}$ are trainable parameters in the transformation phase. Finally, we obtained the representation of students $h_{s_i}$, exercises $h_{e_j}$, and concepts $h_{c_z}$. The remaining question is how to derive the Mas and exercises' difficulty levels, and how to predict the performance of students on the exercises.

## 4.2 Interaction Functions

Interaction Functions (IFs) are devised to predict the likelihood of students correctly answering exercises, which can be formulated as

$$
\hat{y}_{ij} = \sigma(\mathcal{F}((\mathbf{Mas}_{s_i} - \mathbf{Diff}_{e_j}) \odot \mathbf{Q}_{e_j})), \quad (7)
$$

where $\hat{y}_{ij} \in [0, 1]$ represents the prediction outcome of $i$-th student practice $j$-th exercise, $\sigma$ typically employs the Sigmoid, $\mathbf{Mas}_{s_i} \in \mathbb{R}^{1 \times Z}$ denotes the inferred Mas of $i$-th student, $\mathbf{Diff}_{e_j} \in \mathbb{R}^{1 \times Z}$ denotes the inferred difficulty of $j$-th exercise. $\mathbf{Q}_{e_j} \in \mathbb{R}^{1 \times Z}$ signifies the concepts associated with the $j$-th exercise. ICDM is versatile and adaptable to various IFs. For instance, it is compatible with traditional IFs like MIRT that utilizes the logistic function for $\mathcal{F}(\cdot)$, where $\mathbf{Mas}_{s_i} \equiv h_{s_i}, \mathbf{Diff}_{e_j} \equiv h_{e_j}$. Additionally, ICDM can also support recent approaches like NCDM [25], which employs MLPs for $\mathcal{F}(\cdot)$, where $\mathbf{Mas}_{s_i} \equiv h_{s_i}, \mathbf{Diff}_{e_j} \equiv h_{e_j}$, all the while ensuring the weights remain non-negative to uphold the monotonicity assumption. However, in practical learning environments, when assessing whether a student can solve a particular exercise, we frequently base our judgment on the difficulty of other exercises they have tackled previously or Mas of students who have attempted the same exercise (i.e., global-level information). Obviously, previous IFs have neglected this aspect.

**Global-Level Interaction Function.** The global-level information can be effectively captured by introducing layers of Graph Convolutional Network (GCN). Specifically, we adopt the propagation rule from LightGCN [11] due to its proven efficacy in scenarios dominated by ID-features. To facilitate this, we construct a bipartite graph $\mathcal{G}_{se} = (\mathcal{V}_{se}, \mathcal{E}_{se})$ where $\mathcal{V}_{se} = S \cup E$ and $\mathcal{E}_{se}$ involve all observed interactions between students and exercises. The proposed

GLIF can be expressed as

$$\mathbf{Con}_{e_j} = \frac{1}{|\mathcal{RC}_{e_j}|} \sum_{c_z \in \mathcal{RC}_{e_j}} h_{c_z},$$

$$\mathbf{Mas}_{s_i} = \frac{1}{2}(\mathbf{h}_{s_i} + \sum_{e_j \in \mathcal{N}_{s_i}} \frac{1}{\sqrt{|\mathcal{N}_{s_i}||\mathcal{N}_{e_j}|}} \mathbf{h}_{e_j}) \odot \mathbf{Con}_{e_j}, \quad (8)$$

$$\mathbf{Diff}_{e_j} = \frac{1}{2}(\mathbf{h}_{e_j} + \sum_{s_i \in \mathcal{N}_{e_j}} \frac{1}{\sqrt{|\mathcal{N}_{e_j}||\mathcal{N}_{s_i}|}} \mathbf{h}_{s_i}) \odot \mathbf{Con}_{e_j},$$

where $\mathbf{Con}_{e_j} \in \mathbb{R}^{1 \times d}$ denotes the average related concepts' information of the $j$-th exercise, $\mathcal{RC}_{e_j}$ represents the set of related concepts for the $j$-th exercise. $\mathcal{N}_{s_i}$ denotes the neighbors of the $i$-th student in $\mathcal{G}_{se}$, $\mathcal{N}_{e_j}$ denotes the neighbors of the $j$-th exercise in $\mathcal{G}_{se}$. In the end, akin to NCDM, we employ MLPs for $\mathcal{F}(\cdot)$ and use the ReLU to ensure non-negative weights, thereby fulfilling the monotonicity assumption. The prediction is calculated as Eq (7).

## 4.3 Model Training

In the CD task, the main loss function utilized is the binary cross-entropy loss. This computes the discrepancy between the model's predicted outcomes and the actual response scores within a mini-batch. Additionally, we employ regularization term $\Omega(\cdot)$ to mitigate overfitting. The whole loss function can be formulated as

$$\mathcal{L}_{\text{BCE}} = -\sum_{i,j,r_{ij} \in T}^{|T|} r_{ij} \log \hat{y}_{ij} + (1 - r_{ij}) \log(1 - \hat{y}_{ij}), \quad (9)$$

$$\mathbf{H}^{(0)} = H_s \oplus H_r \oplus H_w \oplus H_c, \mathcal{L} = \mathcal{L}_{\text{BCE}} + \lambda_{\text{reg}} \Omega(\mathbf{H}^{(0)}),$$

where $\lambda_{\text{reg}}$ is a hyperparameter to control the weight of regularzation loss in $\mathcal{L}$. $\mathbf{H}^{(0)} \in \mathbb{R}^{(N^O + 2M + Z) \times d}$, $\oplus$ is a concatenation operator. The code is supplied in the supplementary material. $\Omega(\mathbf{H}^{(0)}) = \frac{\|\mathbf{H}^{(0)}\|_{2,2}}{N^O + M}$, where $\|\mathbf{H}^{(0)}\|_{2,2} = \sum_u \sum_v |\mathbf{H}_{uv}^{(0)}|^2$ is an entry-wise matrix norm. We analyze the time complexity of ICDM, and the details are presented in Appendix A.

## 5 EXPERIMENTS

In this section, we first delineate four real-world datasets and evaluation metrics. Then through comprehensive experiments, we aim to manifest the preeminence of ICDM in both transductive and inductive scenarios. To ensure reproducibility and robustness, all experiments are conducted ten times. Our code is available at https://anonymous.4open.science/r/ICDM.

**Table 1: Statistics of real-world datasets for experiments.**

| Datasets | FrcSub | EdNet-1 | Assist17 | NeurIPS20 |
|---|---|---|---|---|
| #Students | 536 | 1776 | 1709 | 2840 |
| #Exercises | 20 | 11925 | 3162 | 6000 |
| #knowledge Concepts | 8 | 189 | 102 | 268 |
| #Response Logs | 10,720 | 616,193 | 390,311 | 214,328 |
| Sparsity | 1.0 | 0.029 | 0.072 | 0.012 |
| Average Correct Rate | 0.530 | 0.662 | 0.815 | 0.631 |
| **Q** Density | 2.80 | 2.25 | 1.22 | 4.14 |

## 5.1 Experimental Settings

**Datasets Description.** The experiments are conducted using four real-world datasets: FrcSub, EdNet-1, Assist17, and NeurIPS20. Frc-Sub [4, 21] consists of middle school students' scores on fraction subtraction objective problems. EdNet-1 [2] is the dataset of all student-system interactions collected over 2 years by Santa, a multi-platform AI web-based tutoring service with more than 780K users in Korea. The Assist17 datasets is provided by the ASSISTment web-based online tutoring platforms [5] and are widely used for cognitive diagnosis tasks [25]. The NeurIPS20 dataset is derived from a competition called The NeurIPS 2020 Education Challenge [28]. It contains students' response logs to mathematics questions over two school years (2018-2020) from Eedi, a leading educational platform which millions of students interact with daily around the globe. For more detailed statistics on these four datasets, please refer to Table 1. Notably, "Sparsity" refers to the sparsity of the dataset, which is calculated as $\frac{|T|}{|S||E|}$. "Average Correct Rate" represents the average score of students on exercises, and "**Q** Density" indicates the average number of knowledge concepts per exercise.

**Evaluation Metrics.** To assess the efficacy of ICDM, we utilize both score prediction and interpretability metrics. This approach offers a holistic evaluation from both the predictive accuracy and interpretability standpoints.

Score Prediction Metrics: Evaluating the efficacy of CDMs poses difficulties owing to the absence of the true **Mas**. A prevalent workaround is to appraise these models based on their capability to predict students' scores on exercises in the test data. In line with prior CDM studies [25], in the transductive scenario shown in Figure 4(a), we partition the data into train and test data and assess our model's performance on the test data using classification and regression metrics such as AUC, Accuracy (ACC), and RMSE. The test size is set to 0.2, following the previous researches [14, 25]. Crurcially, we build the SCG solely based on the train data. In the inductive setting depicted in Figure 4(b), we retain the test data intact and partition the training data by students at a ratio $p_n = 0.2$. In this approach, we can obtain two sets of students: $S^O$ and $S^U$. Furthermore, we construct the SCG using only the training data from $S^O$. We then use the training data from $S^U$ to infer $\mathbf{Mas}^U$. Ultimately, accuracy is computed only by the prediction of $S^U$ performance on test data exercises, denoted as $\text{ACC}^{\dagger}$.

Interpretability Metric: Diagnostic results are highly interpretable hold significant importance in CD. In this regard, we employ the degree of agreement (DOA), which is consistent with the approach used in [14, 25]. The underlying intuition here is that, if $s_a$ has a greater accuracy in answering exercises related to $c_k$ than student $s_b$, then the probability of $s_a$ mastering $c_k$ should be greater than that of $s_b$. Namely, $\mathbf{Mas}_{s_a,c_k} > \mathbf{Mas}_{s_b,c_k}$. DOA is defined as Eq. (10)

$$\text{DOA}_k = \frac{\sum\limits_{a,b \in S} \delta\left(\mathbf{Mas}_{s_a,c_k}, \mathbf{Mas}_{s_b,c_k}\right) \frac{\sum_{j=1}^{M} \mathbf{Q}_{jk} \wedge \varphi(j,a,b) \wedge \delta(r_{aj}, r_{bj})}{\sum_{j=1}^{M} \mathbf{Q}_{jk} \wedge \varphi(j,a,b) \wedge I(r_{aj} \neq r_{bj})}}{Z}, \quad (10)$$

where $Z = \sum_{a,b \in S} \delta(\mathbf{Mas}_{s_a,c_k}, \mathbf{Mas}_{s_b,c_k})$, $\mathbf{Q}_{jk}$ indicates exercise $e_j$'s relevance to concept $c_k$, $\varphi(j, a, b)$ checks if both students $s_a$ and $s_b$ answered $e_j$, $r_{aj}$ represents the response of $s_a$ to $e_j$, and $I(r_{aj} \neq r_{bj})$ verifies if their responses are different, $\delta(r_{aj}, r_{bj})$ is 1 for a right response by $s_a$ and a wrong response by $s_b$, and 0 otherwise. Consistent with [14], we compute the average DOA for

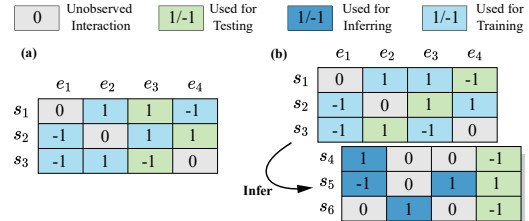

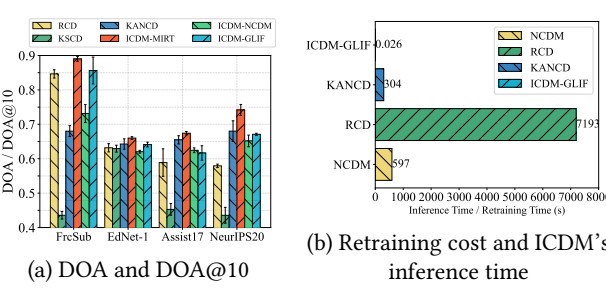

**Figure 4: Different evaluations between transductive CD and inductive CD: (a) Transductive CD. (b) Inductive CD.**

all concepts in FrcSub and the top 10 concepts with the highest number of response logs in EdNet-1, Assist17, NeurIPS20 and refer to it as DOA@10. In the transductive scenario, both DOA and DOA@10 are computed for all students $S$, while in the inductive scenario, they are specifically calculated for new students $S^U$.

**Hyperparameter Settings.** For parameter initialization, we employ the Xavier [7], and for optimization purposes, Adam [13] is adopted. The embedding size is set as 64. The batch size is set as 1024 for EdNet-1, 16 for FrcSub and 128 for other datasets. The k-hops $k$ is tuned from the range $\{1, 2, 3, 4\}$. To regulate the impact of the regularization term, we adjust $\lambda_{\text{reg}}$ within the range $\{10^{-4}, 10^{-3}, \ldots, 1\}$. We set the dropout parameters $\alpha$ as 0.1, $\beta$ as 0.2. Details regarding the experiment can be found in Appendix B.

**(a) DOA and DOA@10**

**(b) Retraining cost and ICDM's inference time**

**Figure 5: Interpretability and running time results.**

## 5.2 Transductive Cognitive Diagnosis

**Baselines and State-of-the-Art Methods.** We compare ICDM against various leading baselines and state-of-the-arts Methods in CD, utilizing the hyperparameter settings and IFs as described in their respective original publications.

• DINA [3] is a traditional CDM which utilize discrete mastery pattern (0 or 1) to model Mas.

• MIRT [20] is a representative model of latent factor CDMs, which uses multidimensional $\theta$ to model Mas.

• NCDM [25] is a recent deep-learning based CDM which utilize MLPs to replace the traditional manually designed IFs.

• RCD [6] leverages GAT to explore the relations among students, exercises and knowledge concepts. Here, to ensure a fair comparison, we solely utilize the student-exercise-concept component of RCD, excluding the dependency on knowledge concepts.

• KSCD [17] also delves into the implicit relationships among knowledge concepts and employs a concept-augmented IF.

• KANCD [26] is an enhanced version of NCDM, delving into the implicit relationships among concepts to tackle the knowledge coverage issue.

To further demonstrate the versatility of ICDM in accommodating various IFs, we not only employ the proposed GLIF but also utilize the traditional MIRT and the classic deep-learniong based IF (i.e., NCDM). These variations are denoted as ICDM-MIRT, ICDM-NCDM, and ICDM-GLIF, respectively.

**Results.** The comparison results are listed in Table 2 and Figure 5(a). When comparing interpretability performance, we particularly opt for KSCD, KANCD, and RCD as baselines. This is because MIRT doesn't support DOA computation due to its latent mastery pattern, while DINA demonstrates subpar prediction performance. Moreover, NCDM has shown a lackluster DOA as highlighted in [26]. We can conclude the following observations:

• ICDM-MIRT and ICDM-NCDM outperform MIRT and NCDM, highlighting the significant impact of ICDM. Moreover, with the advantage of ICDM, the traditional IF MIRT demonstrates competitive predictive performance and the best interpretability performance.

• Despite ICDM-GLIF being primarily tailored for the inductive scenario in CD, it consistently outperforms the current state-of-the-art CDMs in prediction accuracy. Moreover, it demonstrates commendable interpretability performance across all four datasets.

## 5.3 Inductive Cognitive Diagnosis

**Baselines and Compared Methods.** We conduct a comparison of ICDM against other baselines and utilize the hyperparameter settings and IFs described in their respective original publications.

• Random: It predicts students' scores based on a uniform distribution ranging from 0 to 1.

• KANCD-Mean: We incorporate the postprocessing mean strategy from IMCGAE into KANCD, as the original KANCD was designed solely for the transductive scenario. It assigns the embedding of new students to the average of the old students.

• KANCD-Closest: For each new student in $S^U$, we assign their embedding based on the most similar student in $S^O$, who is selected based on the similarity of response logs.

• IMCGAE [19]: It utilizes a graph autoencoder combined with a postprocessing strategy tailored for inductive rating prediction. We select IMCGAE as the representative because INMO cannot be applied in CD, and the performance of IMCGAE in [19, 31] is superior to previous methods like IGMC [34], IDCF [30]. We modify it accordingly to suit the inductive CD task.

• KANCD-Re: By integrating the train data of $S^U$ into the training phase, we retrain KANCD.

• ICDM-Re: By incorporating the train data of $S^U$ into the training phase, we retrain ICDM.

We chose to compare with KANCD because it exhibits outstanding performance in the transductive scenario. To further demonstrate the performance of ICDM, we compare with ICDM-Re and KANCD-Re. These represent the performance of ICDM and KANCD after retraining, evaluating their performance on the test data of $S^U$. These can be seen as an upper bound for the inductive scenario. The aggregator of ICDM is selected as the mean operator due to its high efficiency and outstanding performance, which will be demonstrated in subsequent sections.

**Table 2: Overall prediction performance in transductive scenario. In each column, an entry with the best mean value is marked in bold and underline for the runner-up. The standard deviation is not shown in the table since it is very low (less than 0.001).**

| Datasets | FrcSub | | | EdNet-1 | | | Assist17 | | | NeurIPS20 | | |
|---|---|---|---|---|---|---|---|---|---|---|---|---|
| Algo. | AUC | ACC | RMSE | AUC | ACC | RMSE | AUC | ACC | RMSE | AUC | ACC | RMSE |
| DINA | 0.6623 | 0.5547 | 0.5436 | 0.5305 | 0.4245 | 0.5651 | 0.6777 | 0.5339 | 0.4726 | 0.6324 | 0.4372 | 0.5805 |
| MIRT | 0.8609 | 0.7890 | 0.3903 | 0.6668 | 0.6733 | 0.4679 | 0.8804 | 0.8594 | 0.3138 | 0.6849 | 0.6570 | 0.4889 |
| NCDM | 0.7756 | 0.5264 | 0.4994 | 0.7419 | 0.7082 | 0.4362 | 0.8727 | 0.8491 | 0.3201 | 0.7775 | 0.7178 | 0.4302 |
| RCD | 0.8601 | 0.7848 | 0.4058 | 0.7719 | 0.7333 | 0.4214 | 0.8926 | 0.8663 | 0.3051 | 0.7713 | 0.7156 | 0.4313 |
| KSCD | 0.8956 | 0.8227 | 0.3573 | 0.7476 | 0.6961 | 0.4347 | 0.8950 | 0.8653 | 0.3048 | 0.7699 | 0.7135 | 0.4326 |
| KANCD | 0.9031 | 0.8386 | 0.3515 | 0.7553 | 0.7226 | 0.4287 | 0.8900 | 0.8621 | 0.3082 | 0.7662 | 0.7150 | 0.4315 |
| ICDM-MIRT | 0.8934 | 0.8195 | 0.3606 | 0.7523 | 0.7226 | 0.4294 | 0.8960 | 0.8682 | 0.3031 | 0.7712 | 0.7175 | 0.4308 |
| ICDM-NCDM | 0.9026 | 0.8331 | 0.3517 | 0.7535 | 0.7217 | 0.4297 | 0.8911 | 0.8635 | 0.3067 | 0.7794 | 0.7232 | 0.4278 |
| ICDM-GLIF | **0.9053** | **0.8397** | **0.3496** | **0.7565** | **0.7234** | **0.4284** | **0.8979** | **0.8705** | **0.3013** | **0.7796** | **0.7233** | **0.4276** |

**Table 3: Overall prediction performance in inductive scenario. The symbol "*" indicates the retraining results. Details are as same as Table 2.**

| Dataset | Metric | Random | KANCD-Mean | KANCD-Closest | KANCD-Re* | IMCGAE (MIRT) | ICDM (MIRT) | ICDM-Re | IMCGAE (NCDM) | ICDM (NCDM) | ICDM-Re | IMCGAE (GLIF) | ICDM (GLIF) | ICDM-Re* |
|---|---|---|---|---|---|---|---|---|---|---|---|---|---|---|
| FrcSub | ACC† | 0.5083 | 0.6234 | 0.7329 | 0.8325* | 0.6130 | **0.7373** | 0.8221 | 0.5988 | **0.7193** | 0.8346 | 0.6130 | **0.7188** | 0.8352* |
| | DOA | \ | 0.4882 | 0.6145 | 0.8058* | 0.5128 | **0.8220** | 0.8819 | 0.5133 | **0.6662** | 0.7086 | 0.5128 | **0.7140** | 0.8127* |
| EdNet-1 | ACC† | 0.5009 | 0.6229 | 0.6951 | 0.7259* | 0.6989 | **0.7013** | 0.7270 | 0.6797 | **0.6994** | 0.7238 | 0.7026 | **0.7036** | 0.7263* |
| | DOA | \ | 0.5523 | 0.5149 | 0.6466* | 0.5097 | **0.6018** | 0.6690 | 0.5412 | **0.5900** | 0.6315 | 0.5890 | **0.6033** | 0.6545* |
| Assist17 | ACC† | 0.4993 | 0.8486 | 0.8413 | 0.8633* | 0.8499 | **0.8552** | 0.8677 | 0.8241 | **0.8576** | 0.8623 | 0.8598 | **0.8620** | 0.8691* |
| | DOA | \ | 0.4906 | 0.5520 | 0.6375* | 0.6136 | **0.6215** | 0.6758 | 0.5529 | **0.5915** | 0.6287 | 0.6506 | **0.6506** | 0.5823* |
| NeurIPS20 | ACC† | 0.5001 | 0.6445 | 0.5951 | 0.7169* | 0.6451 | **0.6568** | 0.7161 | 0.6408 | **0.6552** | 0.7195 | 0.6572 | **0.6567** | 0.7199* |
| | DOA | \ | 0.5159 | 0.5560 | 0.7258* | 0.5623 | **0.5957** | 0.7443 | 0.5183 | **0.5444** | 0.6375 | 0.5437 | **0.5478** | 0.6492* |

**Table 4: Overall prediction performance of ablation study in inductive scenario for EdNet-1. Details are as same as Table 2.**

| Dataset | EdNet-1 | | | | | |
|---|---|---|---|---|---|---|
| IF | MIRT | | NCDM | | GLIF | |
| Metric | ACC | DOA | ACC | DOA | ACC | DOA |
| ICDM-w.o.C | 0.7012 | 0.5913 | 0.6980 | 0.5739 | 0.7032 | 0.6012 |
| ICDM-w.o.drop | 0.7005 | 0.5958 | 0.6994 | 0.5757 | 0.7008 | 0.6003 |
| ICDM-w.o.tf | 0.6885 | 0.5154 | 0.6916 | 0.5460 | 0.6906 | 0.5326 |
| ICDM | **0.7013** | **0.6018** | **0.6994** | **0.5900** | **0.7036** | **0.6033** |

**Results.** The comparison results are listed in Table 3. We can conclude the following key observations:

• Transductive CDMs with a simple postprocessing mean strategy perform better than RANDOM. However, they still don't produce satisfactory results and fall significantly short compared to the outcomes achieved after time-consuming retraining. This indicates that the inductive scenario in CD is not a trivial task, and merely using previous transductive CDMs is not feasible.

• ICDM consistently outperforms the IMCGAE on almost all datasets regardless of the IF used. This demonstrates that ICDM is more effective than IMCGAE under the inductive scenario in CD.

• ICDM-Re outperforms KANCD-Re in most cases, further demonstrating the superior performance of ICDM. We are pleasantly surprised to find that, for instance, the accuracy of ICDM-GLIF can reach up to 86.06% for FrcSub, 96.87% for EdNet-1, 96.74% for Assist17, and 91.22% for nips20 after retraining ICDM. This is ample evidence to prove the efficacy of ICDM to provide immediate feedback for new students in WOIDSs.

**Inference Time Comparison.** In WOIDSs, ICDM's ability to circumvent the constant need for retraining CDMs is highly advantageous. Here, we show the speed at which ICDM infers the Mas of new students on EdNet-1, compared with the difference of transductive CDMs that infer via retraining. As illustrated in Figure 5(b), retraining transductive CDMs requires significantly more time, which evidently cannot meet the rapid feedback demands for new students in WOIDSs. Conversely, ICDM efficiently derives the Mastery Level (Mas) of 355 new students with 112,535 response logs in the EdNet-1 in **26ms**, while retaining **96.87%** ACC of retraining.

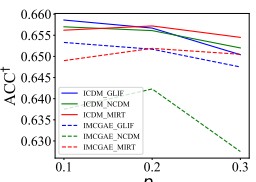

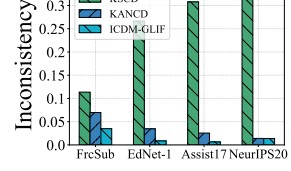

(a) Performance under different $p_n$ in EdNet-1

(b) Consistency of inferred Mas

**Figure 6: Prediction performance and consistency results.**

**Ablation Study.** To showcase the contributions of each component in ICDM, we conduct an ablation study on ICDM, which is divided into the following three versions. ICDM-w.o.-C: This version removes the potential "desired" relationship between students and concepts in the SCG. ICDM-w.o.-Drop: This version removes the layer-wise neighbor dropout in the CAGT process. ICDM-w.o.-tf: This version removes the transformation phase within the CAGT

process. Specifically, the embeddings' dimension is set to $Z$. Due to space limitations, we only present the ablation study on the EdNet-1. Ablation studies on other datasets can be found in Appendix B. As shown in Table 4, ICDM surpasses other versions in both prediction and interpretability performance. This suggests that these components, when combined, enhance ICDM. When each component is removed individually, either the prediction performance decreases or the interpretability performance suffers, indicating that each component plays a crucial role.

**The Effect of** $p_n$**.** In real educational scenarios, there might be a large influx of students practicing in WOIDSs within a short period (e.g., large-scale unified online exams), which means a high $p_n$. Therefore, it is necessary to evaluate the relationship between the model's performance and $p_n$. Due to space limitations, we only present the outcome of EdNet-1, other datasets can be found in Appendix C. In Figure 6(a), ICDM exhibits a higher ACC$^{\dagger}$ compared to IMCGAE, especially as $p_n$ increases. This demonstrates the robustness of ICDM in handling larger batches of new students, underscoring its superiority in inductive scenarios for WIODSs.

### 5.4 Analysis of Diagnosis Results (i.e., Mas)

**The Distrubution of Students' Mas.** Indeed, students can naturally be grouped into categories based on their performance, such as those with low and high correct rates. This classification reflects intrinsic differences in their Mas. We employ t-SNE [23], a renowned dimensionality reduction method, to map the **Mas** onto a two-dimensional plane. By shading the scatter plot according to the corresponding correct rates, with deeper shades of blue indicating higher correct rates, we achieve a visual representation of the students' Mas distribution. From Figure 7(a) and Figure 7(b), it is clear that ICDM-GLIF clusters all students $S$ with high accuracy rates more cohesively than KANCD. Moreover, ICDM-GLIF-Ind denotes the version of ICDM infer Mas of new students $S^U$ without retraining. As depicted in Figure 7(c), the inferred Mas of new students is plausible, as new students with similar correct rates (colored in green) cluster closely with older students (colored in blue) of comparable rates.

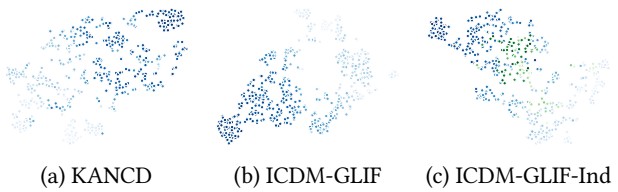

(a) KANCD     (b) ICDM-GLIF     (c) ICDM-GLIF-Ind

**Figure 7: Visualizations of inferred Mas on FrcSub dataset.**

**The Consistency of Mas.** Previous CDMs overlook the consistency of inferred Mas. There are instances where two students have very similar or even identical response logs, yet their Mas differ. This is unfair for both students and can adversely affect other recommendation algorithms within WOIDSs. To uncover this phenomenon, we introduce a metric termed "Inconsistency". Firstly, for each student $s_i$, we utilize cosine similarity to select another student $s_j$ whose response logs are the most similar to $s_i$ which is

calculated as $s_j = \arg\max_{s_j \in S \setminus s_i} \text{Cosine}(\mathbf{R}_i, \mathbf{R}_j)$. Then, we calculate "Inconsistency" as $\frac{1}{Z} \frac{1}{N} \sum_{i=1}^{N} \|\mathbf{Mas}_{s_i} - \mathbf{Mas}_{s_j}\|_1$. As depicted in Figure 6(b), we have chosen KANCD and KSCD for comparison because they represent the most recent CDMs that exhibit competitive performance. It is evident that the inconsistency of ICDM-GLIF is significantly lower than that of KANCD and KSCD. This indicates that ICDM-GLIF's inferred Mas exhibits higher consistency, making it more suitable for downstream algorithms in WOIDSs.

### 5.5 Hyperparameters Analysis

**The Effect of Different Aggregators.** We conduct experiments to demonstrate the impact of different aggregators on ICDM-GLIF, as illustrated in Figure 8(a). Using GATV2 resulted in out-of-memory for the last three datasets, hence it is not displayed in the figure. The mean and pool aggregators surpasse other attention-based aggregators (e.g., GAT, GATV2). This can be attributed to the fact that CD's features are solely based on IDs. Employing complex non-linear transformations like attention might not lead to performance improvement and could even result in a decline, as also mentioned in [11, 29]. The mean operator is recommended as it exhibits stable and relatively good performance across all four datasets.

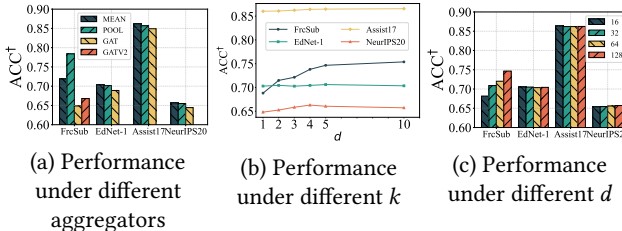

(a) Performance under different aggregators    (b) Performance under different $k$    (c) Performance under different $d$

**Figure 8: Hyperparameters analysis results.**

**The Effect of** $k$**.** $k$ determines the receptive field size of ICDM-GLIF in the SCG. As shown in Figure 8(b), as $k$ increases, ACC also increases, but the computational time grows correspondingly. The relationship between computation time and $k$ will be shown in Appendix D. High computational time is contrary to our intention of providing immediate diagnostic results for new students in WOIDSs. We recommend using $k = 3$ or $k = 4$, which offers decent performance in a relatively shorter time.

**The Effect of** $d$**.** $d$ controls the dimension of the embedding. As shown in Figure 8(c), the preferable choices for $d$ are 64 or 128 due to their stable performance across all four datasets.

### 6 CONCLUSION

This paper proposes an inductive cognitive diagnosis model (ICDM) for fast new students' mastery levels inference in WOIESs. ICDM mainly focuses on handling inductive scenario in CD which can provide immediate feedback for new students. A construction-aggregation-generation-transformation process is introduced to extract features effectively from the newly proposed student-centered graph. ICDM still has some limitations. It cannot infer the difficulty or discrimination of new incoming exercises. We look forward to finding a solution that can address this issue reasonably and effectively which is also of vital importance to WOIDSs.

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

# APPENDIX

The appendix is organized as follows:

• Appendix A analyzes the ICDM's time complexity and compares it with RCD.

• Appendix B presents the detailed settings of compared baselines and other details about experiments.

• Appendix C further supplements the analysis with additional details regarding the hyperparameter analysis.

## A  TIME COMPLEXITY ANALYSIS

In this section, we present a detailed time complexity analysis of our proposed model ICDM. We compare our time complexity with that of RCD, as RCD is the only CDM based on Graph Neural Networks.

**Time Complexity Analysis of ICDM.** In ICDM, we construct a student-centered graph (SCG) $\mathcal{G}$ with four node and edge types based on $\mathbf{R}^O$, $\mathbf{I}^O$ and $\mathbf{Q}$. We choose the mean operator as the aggregator in the CAGT process for illustration purposes. Given that we do not employ the non-linear activation and feature transformation usually found in GNNs, the time complexity can be straightforwardly computed as $O(2(\|\mathbf{Q}_B\|_0 + \|\mathbf{R}^O_B\|_0 + \|\mathbf{I}^O_B\|_0)kd + 5d^2 + 3Zd)$ for the CAGT process, where $B$ denotes the pre-defined batch size and $\mathbf{Q}_B$ refers to the entries in $\mathbf{Q}$ that are related to the students, exercises, and concepts present in that batch. Similarly, $\mathbf{R}^O_B$ and $\mathbf{I}^O_B$ follow the same logic. $\|\mathbf{Q}_B\|_0, \|\mathbf{R}^O_B\|_0, \|\mathbf{I}^O_B\|_0$ represents non-zero number of $\mathbf{Q}_B, \mathbf{R}^O_B, \mathbf{I}^O_B$ respectively. $k$ represents the k-hops. $d$ stands for the size of the embeddings, $Z$ denotes the number of knowledge concepts. The most time-consuming part is the aggregation step which takes $O(2(\|\mathbf{Q}_B\|_0 + \|\mathbf{R}^O_B\|_0 + \|\mathbf{I}^O_B\|_0)kd)$. $L$ denotes the number of GAT layers.

**Time Complexity Analysis of RCD.** In RCD, an exercise-concept graph is constructed using $\mathbf{Q}$ and a student-exercise graph is formed using $\mathbf{R}$. Given that RCD employs the graph attention network, which necessitates the computation of attention coefficients

between every pair of connected nodes, its time complexity belongs to $O(2(\|\mathbf{R}\|_0 + \|\mathbf{Q}\|_0)LZ^2)$. Herein, $Z$ represents the number of concepts ($d \ll Z$).

ICDM evidently takes less time compared to RCD due to two main reasons. First, in each batch during training, we only need to extract the relevant part from the constructed graph to perform graph convolution, while RCD needs to perform graph convolution on the entire graph. Second, in the CAGT process, the transformation phase reduces the embedding dimension to $d$, where $d$ is much smaller than $Z$.

## B  EXPERIMENTAL DETAILS

**Implementation Details** This section delineates the detailed settings when comparing our method with the baselines and state-of-the-art methods in both transductive scenario and inductive scenario. All experiments are run on a Linux server with two 3.00GHz Intel Xeon Gold 6354 CPUs and one RTX3090 GPU. All the models are implemented by PyTorch [18].

**Transductive Scenario.** In the following section, we will elaborate on some details regarding the utilization of compared methods.

• DINA [3] is a representative CDM which models the mastery pattern with discrete variables (0 or 1).

• MIRT [20] is a representative model of latent factor CDMs, which uses multidimensional $\boldsymbol{\theta}$ to model the latent abilities. We set the latent dimension as 16 which is the same as [25]

• NCDM [25] is a deep learning based CDM which uses MLPs to replace the traditional interaction function (i.e., logistic function). We adopt the default parameters which are reported in that paper.

• RCD [6] leverages GNN to explore the relations among students, exercises and knowledge concepts. Here, to ensure a fair comparison, we solely utilize the student-exercise-concept component of RCD, excluding the dependency on knowledge concepts.

• KANCD [26] improves NCDM by exploring the implicit association among knowledge concepts to address the problem of knowledge coverage. Here, we adopt the default parameters which are reported in that paper.

• KSCD [17] also explores the implicit association among knowledge concepts and leverages a knowledge-enhanced interaction function. Due to the absence of open-source code online, we have independently replicated KSCD.

The implementation of DINA, MIRT, NCDM and KANCD comes from the public repository https://github.com/bigdata-ustc/EduCDM. For RCD, we adopt the implementation from the authors in https://github.com/bigdata-ustc/RCD.

**Inductive Scenario.** IMCGAE [19] utilizes a graph autoencoder combined with a postprocessing strategy tailored for inductive rating prediction. We select IMCGAE as the representative because INMO cannot be applied in CD, and the performance of IMCGAE in [19, 31] is superior to previous methods like IGMC [34], IDCF [30]. We modify it accordingly to suit the inductive CD task. Specifically, we construct the necessary student-exercise bipartite graph for IMCGAE and replace its initial bilinear decoder with IFs in CD. The implementation of IMCGAE [19] comes from the version of https://github.com/WuYunfan/igcn_cf/.

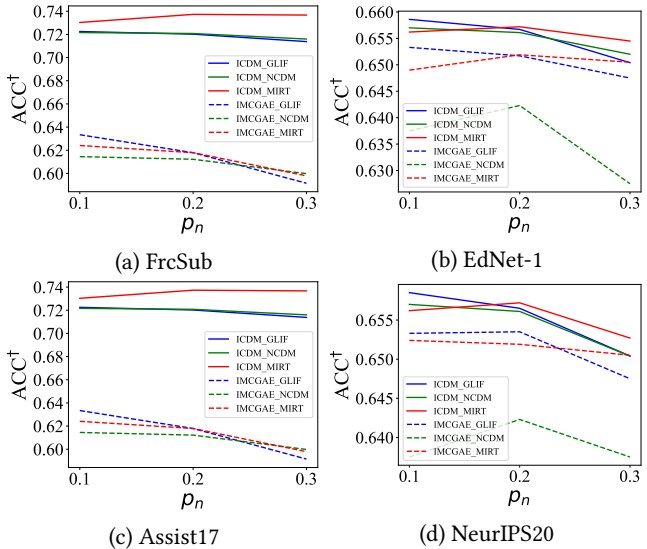

(a) FrcSub

(b) EdNet-1

(c) Assist17

(d) NeurIPS20

**Figure 9: Performance under different $k$.**

**Table 5: Overall performance of ablation study in inductive scenario for the remaining datasets. Details are the same as Table 2.**

| Dataset | FrcSub | | | | | | Assist17 | | | | | | NeurIPS20 | | | | | |
|---|---|---|---|---|---|---|---|---|---|---|---|---|---|---|---|---|---|---|
| IF | MIRT | | NCDM | | GLIF | | MIRT | | NCDM | | GLIF | | MIRT | | NCDM | | GLIF | |
| Metric | ACC | DOA | ACC | DOA | ACC | DOA | ACC | DOA | ACC | DOA | ACC | DOA | ACC | DOA | ACC | DOA | ACC | DOA |
| ICDM-w.o.C | 0.7359 | 0.8060 | 0.7189 | 0.6623 | 0.6998 | 0.6813 | 0.8542 | 0.6060 | 0.8556 | 0.5904 | 0.8589 | 0.5633 | 0.6584 | **0.6222** | 0.6520 | 0.5528 | 0.6518 | 0.5401 |
| ICDM-w.o.drop | 0.7370 | 0.8029 | 0.7168 | 0.6565 | 0.7158 | 0.7011 | 0.8547 | 0.5953 | 0.8576 | 0.5777 | 0.8618 | 0.5416 | 0.6551 | 0.5905 | 0.6513 | **0.5534** | 0.6535 | 0.5471 |
| ICDM-w.o.tf | 0.6574 | 0.6469 | 0.6482 | 0.6134 | 0.6648 | 0.6593 | 0.8495 | 0.4958 | 0.8464 | 0.5910 | 0.8520 | **0.5653** | 0.6561 | 0.5673 | 0.6280 | 0.5224 | 0.6462 | 0.5405 |
| ICDM | **0.7373** | **0.8220** | **0.7193** | **0.6662** | **0.7188** | **0.7140** | **0.8552** | **0.6215** | **0.8576** | **0.5915** | **0.8620** | 0.5349 | **0.6568** | 0.5957 | **0.6552** | 0.5444 | **0.6567** | **0.5478** |

**Ablation Study.** To showcase the contributions of each component in ICDM, we conducted an ablation study on ICDM, which is divided into the following three versions:

• ICDM-w.o.-C: This version removes the potential "desired" relationship between students and concepts in the SCG.

• ICDM-w.o.-Drop: This version removes the layer-wise neighbor dropout in the CAGT process.

• ICDM-w.o.-tf: This version removes the transformation phase within the CAGT process. Specifically, the embeddings' dimension is set to $Z$.

As illustrated in Table 5, ICDM outperforms other versions in terms of both prediction and interpretability performance. This indicates that the amalgamation of these components amplifies the efficacy of ICDM. When components are individually omitted, there is a noted decrease in either predictive accuracy or interpretability performance, showcasing the pivotal role each element holds in the cohesive functionality of the model. Notably, the ICDM-w.o.-tf version exhibits the most substantial decline in performance, emphasizing the significance of the transformation phase within the CAGT process.

**The Effect of $p_n$.** In Figure 9, ICDM consistently demonstrates superior performance over IMCGAE across various IFs and on all four datasets. This superiority becomes more pronounced as $p_n$ grows. Such results underscore the robustness of ICDM, especially when handling increasing numbers of new students, further underscoring its potency and dominance in inductive settings within WOIDSs.

## C HYPERPARAMETER ANALYSIS

**Relationship of $k$ and Computational Time.** The computational time includes both training time and inference time. Training time significantly influences the efficiency of model retraining in WOIDSs. A shorter training time allows for more frequent model updates, facilitating a more accurate and timely estimation of the Mas of new students. As depicted in Figure 10, the training time dramatically increases with the growth of $k$. This shows a significant escalation in computational costs as $k$ becomes larger, indicating a trade-off between the choice of $k$ and the computational efficiency of the model.

The inference time significantly affects the speed at which ICDM can deduce the Mas of new students. A shorter inference time allows for quicker evaluations, enabling the model to promptly provide feedback or adapt learning pathways based on the inferred Mas of the students. This efficiency is crucial in educational settings where immediate feedback and adaptability are essential for enhancing the learning experience and outcomes of new students. As illustrated

in Figure 11, there is a clear trend showing that as $k$ increases, the inference time also rises rapidly.

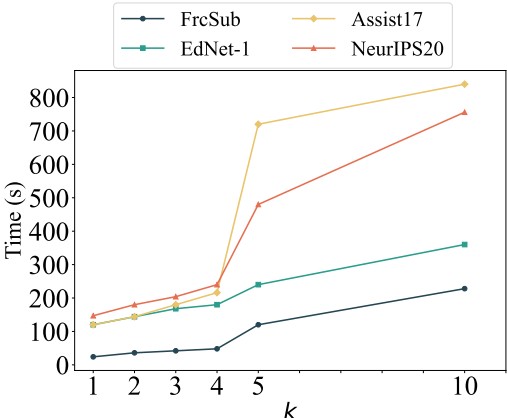

**Figure 10: Relationship of $k$ and training time.**

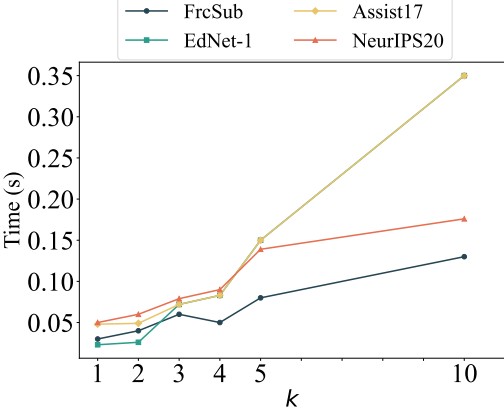

**Figure 11: Relationship of $k$ and inference time.**

