# OpenReview forum: "Inductive Cognitive Diagnosis for Fast Student Learning in Web-Based Intelligent Education Systems"
_ACM.org/TheWebConf/2024/Conference — TheWebConf24_

### Official Review · Reviewer_YVZG · 2023-11-22

**Novelty:** 4
**Technical Quality:** 4

**Review:**

In this paper, the authors propose a novel graph-based method, namely Inductive Cognitive Diagnosis Model (ICDM), to handle the understudied inductive scenario of cognitive diagnosis problem. This work is of decent quality, presented with appropriate clarity. Besides, this study addresses a relatively overlooked yet important area, thus possesses good originality and significance.

Pros:
- Contents of this study are inclusive. Extensive experiments help with demonstrating the efficacy and efficiency of the proposed model in different settings. Studies on hyperparameters, generated student representations and usefulness of components of ICDM are beneficial for readers to have a better understanding of ICDM.
- Informative figures help with illustrating the problem and framework, releasing the readers from toughly comprehending the complicated problem.
- All experiments results are provided in detail, ensuring excellent reproducibility of this work.

Cons:
- For efficacy, the performance enhancement in terms of accuracy seems marginal and lack consistency. The major proposed method, ICDM-GLIF, performs well in the transductive setting but fails in the competition with ICDM-MIRT in the inductive setting, conflicting with the claim the authors make in Line 658 – 661.
- It seems unfair to compare the inference speed of ICDM-GLIF with the retraining time consumption of baselines. As ICDM can also be retrained according to Table 3, the experiments can be revised to compare ICDM with KANCD for inference and compare ICDM with all baselines for retraining.

Typo:
- Line 160: maintains → maintaining

**Questions:**

What does the KANCD stand for in Figure 5(b)? Is it KANCD-re or other variants?

**Reviewer Confidence:**

3: The reviewer is confident but not certain that the evaluation is correct

**Scope:**

4: The work is relevant to the Web and to the track, and is of broad interest to the community

---

### Official Review · Reviewer_hL2w · 2023-11-23

**Novelty:** 4
**Technical Quality:** 3

**Review:**

This paper proposes an Inductive Cognitive Diagnostic Model (ICDM) for fast student learning in Web-Based Online Intelligent Education Systems (WOIESs). The ICDM leverages the Student-Centric Graph (SCG) introduced in this paper to infer students' mastery levels (Mas), enabling immediate feedback to new students without retraining the model. The experiments with real-world datasets show that the ICDM is much faster while maintaining competitive inference performance for new students.

Strong Points

(S1) Novelty
This is the first attempt in WOIESs to tackle a real-world inductive scenario.

(S2) Evaluation Using Real-World Datasets
The paper conducts comparative experiments by using four real-world datasets.

Weak Points

(W1) Unfair Evaluation
The authors claim that, to infer new students' Mas, existing CDM models require retraining from the scratch, totally ignoring the models already trained. However, is it true? Is it impossible to retrain the trained CDM models with the response log of new students? With a small amount of additional time, the models will be refined for the new students.

(W2) Small improvement
The paper claims two main contributions: (Contribution A) Proposing a model using SCG to infer Mas of existing students (let's refer this as Cont-A), and (Contribution B) Adding new students to SCG to infer their Mas without retraining  (let's refer this as Cont-B). However, in terms of accuracy, Cont-A shows a little improvement over KANCD. For Cont-B, the results with ACC are different depending on datasets, compared with KANCD-Closest. Note that KANCD is not a model designed for the inductive scenario; also, Closest is a naive method that simply adopts the embedding of the closest trained students. Thus, the results are not impressive.

(W3) Inappropriate positioning
As mentioned, Cont-A does not seem to be a contribution since its effectiveness is not significant.  Cont-B seems orthogonal to existing models; i.e., it can be applied to other models. If this is the case, is it possible to apply Cont-B to other models and show its effectiveness by comparing the models with and without cont-B?

**Questions:**

Question
1. If there are representations for exercises and concepts in existing CDMs (as shown in Figure 1), it seems possible to infer the Mas of new students by using them. Also, is it possible to apply cont-B to existing CDM models?
2. Is it not possible to retrain existing CDM models with the response log of new students?
3. What does each dimension represent in the representation h_{cz}?
4. Does h_{si} consider the factors like the difficulty of exercises previously solved by the student? Is there a difference between the representation h_{si} of a student who understands all concepts and that of a student who does not understand?
5. In Figure 6 b), why are existing CDMs inconsistent? It is difficult to understand this result. At first, I thought it coud be merely a random initialization issue of parameters; however, we know the models were not retrained for this experiment.

**Reviewer Confidence:**

4: The reviewer is certain that the evaluation is correct and very familiar with the relevant literature

**Scope:**

3: The work is somewhat relevant to the Web and to the track, and is of narrow interest to a sub-community

---

### Official Review · Reviewer_bvpv · 2023-11-29

**Novelty:** 5
**Technical Quality:** 5

**Review:**

This paper proposes a new model to derive cognitive diagnosis in an inductive approach. Unlike traditional methods that infer students' mastery levels by updating individual student embeddings based on their response logs, SCG uses a different strategy. It determines mastery levels by aggregating outcomes from a student's neighbors in the graph. This shift focuses on finding the most appropriate representations for different node types in the SCG, rather than individual student embeddings. The advantage of this method is its efficiency, as it doesn't require retraining for each student-specific case.

This paper aims to identify an important question that serves the online education community and is topic-wise relevant to theWebConf. It aims to address the retraining burden of the state-of-the-art models by using graph approach. It is generally well written, and the model performance is evaluated by an adequate number of experiments. To proceed, it will be great if the following two suggestions could be incorporate.

1. Clarity
While the paper tries to use mathematical notations to clearly define the problem, dataset and model; quite a few of the notations were used without a clear definition, such as in Section 3. This will hinder's the readability since the audience would have to rely on their intuition to interpret the notations.

2. Performance
While the model generates adequate performances in the transductive scenario, it seems to me that the model significantly underperform the retrained version in the inductive scenario, which is probably more important in this context than the transductive one. Depending on the dataset, the difference between the proposed method and the SOTA method can have 2% to 10% performance difference. I would expect more interpretation and justification on why we would outweigh the retraining over such large performance difference.

**Questions:**

While I think the clarity is easier to address in the revision, I would expect more discussion on the performance.

**Reviewer Confidence:**

2: The reviewer is willing to defend the evaluation, but it is likely that the reviewer did not understand parts of the paper

**Scope:**

3: The work is somewhat relevant to the Web and to the track, and is of narrow interest to a sub-community

---

### Official Review · Reviewer_vNc6 · 2023-11-30

**Novelty:** 4
**Technical Quality:** 6

**Review:**

This paper offers an approach to infer quickly new students’ mastery level, as we cannot build a new representation for them given little to no information. A student-centered graph is created and the neighbors of the student are used for their already built representations.

Strong points of the paper:
- Intuitive solution to an important problem within the area.
- The evaluation section is good.
- The code is provided and the datasets used are common.
- Typical evaluation metrics are used, along with an interpretability metric (DOA).

Points that could be improved:
- The idea is not overly novel, even though its application in this problem could be.
- The values for the tested hyperparameters are mentioned, but it is unclear how parameter selection was made.
- Figures 5, 6, and 8 could be a bit bigger, to be more readable.

**Questions:**

N/A

**Reviewer Confidence:**

3: The reviewer is confident but not certain that the evaluation is correct

**Scope:**

4: The work is relevant to the Web and to the track, and is of broad interest to the community

---

### Official Review · Reviewer_45gH · 2023-11-30

**Novelty:** 4
**Technical Quality:** 6

**Review:**

The authors proposed an inductive cognitive diagnosis model to efficiently infer the mastery levels of new students in web-based open information extraction systems. Within it, a student-centered graph is introduced, allowing the shift from updating student-specific embeddings to deriving inductive mastery levels through aggregated outcomes from graph neighbors. This approach represents various node types without requiring retraining, with a global-level inference function is designed to predict students' performance on exercises. Experimental results on real-world datasets show that the proposed model, in contrast to existing transductive cognitive diagnosis models, offers significantly faster inference while maintaining competitive performance for new students.

Strengths
+ Good contextualization with the Web topics, an exemplified in the introduction
+ The problem formalization is clear, with both concepts and formulas
+ Experiments cover extensive datasets and metrics touching on both performance and interpretability

Limitations
- The experimental results show the framework's superiority over several baselines; however, the reported gains, often at the second or third decimal, raise questions about their practical significance in influencing students' learning outcomes. To address this concern, the authors could provide more extensive analysis or contextualization of these gains. Exploring scenarios where even marginal improvements could translate into meaningful learning enhancements would strengthen the argument for the framework's effectiveness.
- The paper's reliance on machine-learning metrics like AUC and RMSE without establishing a clear connection to how these metrics directly contribute to improved learning on the web is a notable gap. To address this, the authors should delve into the educational implications of their chosen metrics. How do improvements in AUC and RMSE correlate with more effective learning experiences for students? Providing this connection would enhance the paper's relevance and applicability to the educational context. Furthermore, the suggestion to conduct an online evaluation with A/B testing is valuable, as it would offer a real-world context for understanding how the proposed framework performs in practical, dynamic learning environments.
- While the paper addresses various datasets and baselines, offering implementation information, the absence of shared source code constitutes a notable impediment for researchers attempting to replicate the study. This challenge is exacerbated by the creation of certain baselines from scratch. To limit this issue, it is crucial for the authors to prioritize the creation and dissemination of a publicly accessible repository containing the source code. Moreover, there is a need for additional clarity regarding the optimization process of the baselines. Detailing how the baselines are optimized would enhance the understanding of their performance and allow for a more informed (fair) comparison with the proposed model.
- As a minor comment, I suggest that the captions of figures and tables should be extended. Certain figures, like Figure 1, are for instance very dense of information and without a proper caption, it is hard to grasp the key messages.

In conclusion, while the experimental results showcase the framework's superiority, concerns about the practical impact of reported gains persist. Addressing this, along with exploring educational implications of machine-learning metrics and considering an online A/B testing evaluation, would enhance the paper's relevance.

**Questions:**

N/A

**Reviewer Confidence:**

4: The reviewer is certain that the evaluation is correct and very familiar with the relevant literature

**Scope:**

3: The work is somewhat relevant to the Web and to the track, and is of narrow interest to a sub-community

---

### Decision · Program_Chairs · 2024-01-22

**Decision:**

Accept

**Comment:**

The paper presents an inductive method for cognitive diagnosis to infer students' mastery levels in online education systems. Reviewers noted that although the presented methodology is not the most novel, they address an important problem in this area. And the comprehensive experiments demonstrate the effectiveness of the proposed approach. The reviewers also indicate that their questions are mostly addressed during the rebuttal. I encourage the authors to carefully revise their paper according to the reviews and rebuttal comments.